# Formation Behavior and Reaction Characteristic of a PTFE/Al Reactive Jet

**DOI:** 10.3390/ma15031268

**Published:** 2022-02-08

**Authors:** Chenghai Su, Huanguo Guo, Yuanfeng Zheng, Jianwen Xie, Haifu Wang

**Affiliations:** State Key Laboratory of Explosion Science and Technology, Beijing Institute of Technology, Beijing 100081, China; 3120205123@bit.edu.cn (C.S.); 7520200026@bit.edi.cn (H.G.); zhengyf@bit.edu.cn (Y.Z.); 3120205130@bit.edu.cn (J.X.)

**Keywords:** PTFE/Al reactive composites, reactive composite jet, shaped charge, formation characteristics, temperature distribution

## Abstract

To reveal the expansion phenomenon and reaction characteristics of an aluminum particle filled polytetrafluoroethylene (PTFE/Al) reactive jet during the forming process, and to control the penetration and explosion coupling damage ability of the reactive jet, the temperature and density distribution of the reactive jet were investigated by combining numerical simulation and experimental study. Based on the platform of AUTODYN-3D code, the Smoothed Particle Hydrodynamics (SPH) algorithm was used to study the evolution behaviors and distribution regularity of the morphology, density, temperature, and velocity field during the formation process of the reactive composite jet. The reaction characteristic in the forming process was revealed by combining the distribution of the high-temperature zone in numerical simulation and the Differential Scanning Calorimeter/Thermo-Gravimetry (DSC/TG) experiment results. The results show that the distribution of the high-temperature zone of the reactive composite jet is mainly concentrated in the jet tip and the axial direction, and the reactive composite jet tip reacts first. Combining the density distribution in the numerical simulation and the pulsed X-ray experimental results, the forming behavior of the reactive composite jet was analyzed. The results show that the reactive composite jet has an obvious expansion effect, accompanied by a significant decrease in the overall density.

## 1. Introduction

Fluoropolymer-based reactive composites are solid energetic materials mixed with reactive metal powder, alloy powder, or intermetallic compound in polymer powder (typically PTFE), which can release chemical energy under high dynamic load and high strain rate conditions. Due to its unique properties, reactive composites have high application value in military and civilian fields [1,2]. The PTFE/Al composite material is a typical reactive material, which is prepared by uniformly filling Al particles in a PTFE matrix, and cold-pressing and sintering. This material has become a benchmark for studying the properties and applications of reactive materials. Recently, for PTFE/Al composites, the formulations and fabrications [3], impact initiation [4], and chemical reaction [5] have been studied. Especially in terms of formulation, by adding metal powders or metal oxides, such as MoO3 [6], Bi2O3 [7], and CuO [8], to PTFE/Al composites, the heat release and gas product amount can be improved. The reactive liner is fabricated by means of uniform mixing, cold pressing, and sintering [9,10]. Under the explosive driving action of the shaped charge, the reactive composite jet formed by the reactive material liner can not only penetrate the target similarly to a traditional metal jet, but more importantly, the reactive jet can activate itself after penetrating the interior of the target and cause a violent explosion/deflagration reaction, releasing a large amount of chemical energy, thereby resulting in a more lethal killing/damaging effect on the inside of the target [11].

Early in 2001, Baker and others developed an RLSC (reactive liner shaped charge), which produced a jet containing energetic materials that released chemical energy during the penetration process [12]. Then, the scaled-up RLSCs were designed and tested against large roadway and bridge pier targets, and the results showed that RLSC was an extremely efficient technology to improve the damage effects [13]. Further studies on concrete targets showed that, compared with the Cu jet, the PTFE/Al reactive composite jet could achieve catastrophic damage, whereas the penetration depth of the funnel-shaped hole on the concrete target was insufficient [14,15]. The behavior of the PTFE/Al RLSC penetrating thick steel plate was also studied, and the experimental results presented that, compared with traditional metal liner-shaped charge, the RLSC can produce a larger perforation diameter, but the penetration depth is lower [16]. Based on the above considerations, research on the RLSC has mainly focused on the damage performance of the reactive composite jet for different targets.

However, few studies have investigated the formation characteristics of the reactive composite jet, especially the density and temperature distribution. The combined response behavior of reactive composites under shock loading is a complex problem, which is a strong function of the loading profile and the initial conditions in the material itself. General investigations have shown that a series of responses occur in the reactive composites ranging from simple deformation and phase changes, to chemical reactions, depending significantly on the particle morphology, initial void volume, and loading stimuli [17]. In fact, the average pressure of the detonation wave acting on the reactive liner exceeds 20 GPa, which is far greater than the threshold activated pressure of the reactive composites [18], and may lead to violent chemical reaction of the reactive composite jet during its formation process. Under impact loading compression or strong dynamic load, the internal temperature of the reactive composites will increase due to plastic deformation, injection, and fracture. The impact temperature is directly affected by the impact speed while the reaction rate and reaction efficiency of the reaction material are obviously controlled by the impact temperature [19]. Baker observed that the high pressure inside the reactive composite jet affects the damage effect of the concrete target [14]. In addition, Lee I. proposed that PTFE decomposes at about 530 °C, and at the same time, micron-sized Al particles and PTFE decomposition products participate in the reaction [20]. Dustin observed [21] a violent exothermic reaction of micron-sized Al with fluorinated gases at 600 °C. Temperature is a key parameter that controls the reaction behavior of PTFE/Al reactive composites.

Different from the traditional metal jet, the expansion phenomenon and reaction mechanism of the reactive jet during the forming process are still unclear. These phenomena will greatly affect its damage ability on armor and concrete targets. Therefore, it is necessary to study the reactive jet forming process and reaction characteristic. Here, we began with the numerical simulation of reactive composite jet formation behaviors, and the density and temperature field changes of the reactive composite jet during the formation process were analyzed. Subsequently, the influence of the explosive types, cone angle, and thickness of the reactive liner on the temperature distribution and tip velocity of the reactive composite jet were investigated. Finally, the formation characteristics and reaction characteristics of the reactive composite jet were verified in DSC/TG and X-ray experiments.

## 2. Formation Behavior of the Reactive Composite Jet

### 2.1. SPH Model and Material Parameters

In the numerical simulation of metal jet formation and penetration behavior, Euler and Euler–Lagrangian coupling algorithms are mainly used for analysis. However, the formation behavior of the reactive composite jet has significant particularity, which is mainly manifested in the radial expansion during the formation process. To better describe the phenomenon, the SPH algorithm is generally used. Previous research has shown that the SPH algorithm has better calculation accuracy than the Euler algorithm [11]. This part mainly uses the AUTODYN-3D platform as an example to introduce the modeling method of the RLSC formation behavior.

The RLSC consists of a reactive liner, case, and high-energy explosive. The typical fluoropolymer-based reactive composite’s PTFE/Al composition ratio is 73.5 wt.% PTFE/26.5 wt.% Al. The explosive is 8701, and the case material is #45 steel. The structure of the shaped charge is designed as a boat tail, which reduces the mass of both the main charge and the whole warhead, and maintains similar jet characteristics. The caliber and length of the charge are 48 and 60 mm, the cone angle of the reactive liner is 50°, the thickness of the case is 5 mm, and the wall thickness of the reactive liner is 0.1 CD. The SPH particle size of all materials is 0.5 mm. All the strength models, EOS, and failure models of the reactive liner, part of the explosives, and the case used in the simulation are listed in Table 1.

Fluoropolymer-based reactive composites are special energetic materials, which have the characteristics of being inert and insensitive under normal conditions and undergo non-self-sustained chemical reaction under high strain rate loading. To describe the reactive composites, the material model must include two parts: one is to describe the mechanical behavior of the reactive composites in the inert stage, and the other is to describe the chemical energy release behavior of the reactive composites in the deflagration reaction stage. In addition, it is also necessary to consider factors, such as the time of the reactive composites’ reaction, which increase the complexity of the research on the formation behavior of the reactive composite jet. To facilitate analysis, it is assumed that the reactive composites do not react chemically during the formation process. The reactive composites are modeled with a shock equation of state. The relation between the velocity *U*_s_ and the particle velocity *u*_p_ can be approximated by [22]:*U*_s_ = *c*_0_ + *Su*_p_(1)
where the Grüneisen parameter, *Γ*, is treated as a constant; *c*_0_ and *S* are based on the date from plate-on-plate impact tests performed on the material. The values for *Γ*, *c*_0_, and *S* in Table 2 are obtained from Taylor [23].

The Johnson–Cook strength model is used to describe the reactive liner material, which can describe the behavior of the material under high strain, high strain rate, and high temperature. The #45 steel is also described by the Johnson–Cook strength model. This material model can be expressed as follows:(2)$$\sigma_{y}=\left[A+B{\left({\bar{\varepsilon }}^{P}\right)}^{n}\right]\left[1+\mathrm{Cln}\left({\dot{\varepsilon }}^{*}\right)\right]\left[1-{\left(\frac{T-T_{room}}{T_{m}-T_{room}}\right)}^{m}\right]$$
where *A*, *B*, *C*, *M*, *N* are material constants; ${\dot{\varepsilon }}^{*}$ is the dimensionless strain rate; and ${\bar{\varepsilon }}^{P}$ is the effective plastic strain. *T* is the surrounding temperature, *T_room_* is the room temperature, and *T_m_* is the melting temperature.

The main charge is the 8701 explosive, and its EOS is expressed by the JWL equation. Table 3 represents the parameters of 8701. The simulation model is shown in Figure 1.

### 2.2. Comparison between Reactive Composite Jet and Cu Jet

Different from traditional metal materials, reactive composites are often formed by cold pressing, and their sound velocity is much lower than that of traditional metal materials. Therefore, during the formation process of the reactive composite jet, the jet will expand and disperse, forming a particle stream composed of high-speed moving micro-elements. The typical formation process of the reactive composite jet at different times is shown in Figure 2a. For comparison with the reactive composite jet, the formation process of the Cu jet is also calculated, and the corresponding formation process is shown in Figure 2b, where *h* is the stand-off.

It can be seen from Figure 2 that under the same charge conditions, the jets formed by the two materials show obvious differences in morphology. As time progresses, the reactive composite jet tip shows obvious expansion and divergence, the diameter of the jet tip increases, and it cannot be agglomerated. The Cu jet continues to stretch and grow with time, and the jet tip becomes thinner, showing good cohesion. This is mainly because under the reaction of detonation, the movement process of the reactive liner element is shown in Figure 3a.

The element on the wall of the reactive liner moves towards the axis with the pressing velocity *v*_1_. When it reaches the axis collision point A, it is divided into the reactive composite jet and slug. The jet moves at a velocity *v*_j_, and the slug moves at a velocity *v*_s_, where *β* is the collapse angle of the element of the liner. The moving coordinate system is established with the collision point A (velocity *v*_2_) as the stagnation point, and then the process of the element of the liner colliding to form the jet and slug can be described as steady flow. According to the Bernoulli equation, the wall element of the reactive liner in the moving coordinate system flows to the collision point at a velocity *v*_3_, and still flows to the jet and slug at a velocity *v*_3_. When *v*_3_ is greater than the material sound velocity *C*_0_ and the supersonic collision forms an attached shock wave *β*_c_ greater than *β*, the jet will not condense. Different from traditional metal materials, since fluoropolymer-based reactive composites are often formed by cold pressing and have certain pores inside them, the sound velocity of the material is much lower than that of traditional metal materials (lower than 2000 m/s) [24]. Therefore, the reactive composite jet will diverge during the formation process and will diverge into many small particles in the radial direction.

Due to the large difference in cohesion between the reactive composite jet and Cu jet, the difference in the jet density is significant. The density distribution of the reactive composite jet and the Cu jet at 2.0 CD is shown in Figure 4. When the jet tip reaches 2.0 CD, the overall density of the reactive composite jet decreases below 2.0 g/cm^3^, and the density at the axis of the jet tip even decreases to approximately 1.6 g/cm^3^.

The curve of the reactive composite jet density at the axis changes with time as shown in Figure 5. Gauss points 1–6 are arranged in sequence from outside to inside along the top axis of the liner. Under the detonation pressure, the axis density of the reactive liner increases rapidly from outside to inside, forming the first density peak. Subsequently, the reactive liner is crushed and deformed toward the axis position under the detonation wave, gradually forming the reactive composite jet, and the density of the top of the reactive liner rises again under the crushing action, forming the second density peak shown in Figure 5. Then, the reactive composite jet tip is formed. During the formation process, the density of the reactive composite jet drops rapidly, and an expansion effect occurs. However, for the Cu jet, due to the good cohesiveness maintained, the density at different positions of the jet tip and the slug are all above 8.0 g/cm^3^. In terms of the velocity gradient, under the condition of the same mass, the reactive liner has a higher tip velocity than the Cu liner due to its lower density while the axial velocity gradient of the reactive composite jet is larger than that of the Cu jet, and the reactive composite jet tip has a larger radial velocity, which leads to the radial expansion effect of the reactive composite jet tip.

### 2.3. Temperature Distribution of the Reactive Composite Jet

Figure 6 shows four typical transient temperature field diagrams during the reactive composite jet formation process. Figure 6a shows that under the action of the detonation product, the reactive composites move to the axis of symmetry and then collide. Meanwhile, the temperature of the collision area near the axis of the jet tip is the highest. At this time, the temperature of the liner is relatively high at the top and low at the bottom. Moreover, it can be found that the temperature of the outer wall of the liner rises first due to its proximity to the explosive, and then the temperature of the inner wall increases significantly when the inner wall collides at the axis. In Figure 6d, the axis collision has ended, the temperature distribution inside the jet and the slug tends to be uniform, and the outside temperature of the two wings is higher, because they are still affected by the detonation products at this time. With the movement of the jet, it enters a free state, thus the pressure and temperature drop gradually. As time progresses, the pressure further decreases while the temperature hardly changes.

The temperature time–history curves of the particles at the axis of the reactive liner are shown in Figure 7. The charge detonates at *t* = 0 μs. Subsequently, the reactive liner heats up under the action of the detonation wave, forming a temperature peak, and the inner wall of the liner is crushed and closed at the axis. The temperature of the inner wall continues to rise under crushing collision, and the entire temperature rise process lasts approximately 5 μs. The temperature of the inner wall reaches the highest peak at 7.5 μs after detonation of approximately 1450 K, and then the particle temperature at the top axis of the liner begins to gradually decrease and stabilize.

The temperature of the particles in the middle of the reactive liner with time is shown in Figure 8a. There are two obvious temperature peaks in the curve. The first temperature peak is mainly caused by the detonation wave acting on the middle of the reactive liner form to increase its temperature. The second temperature peak is caused by a violent collision in the middle of the reactive liner being crushed on the axis, and then the temperature drops and tends to be stable. The temperature of the particles in the bottom of the reactive liner with time is shown in Figure 8b. Different from the top and middle particles of the reactive liner, since the reactive liner bottom cannot be crushed to form the reactive composite jet, the temperature at the bottom only rises at an instant under the action of detonation wave, and then the temperature decreases with time and tends to be stable.

## 3. Influencing Factors of Reactive Composite Jet Formation

### 3.1. The Type of Explosive

The RLSC is mainly composed of reactive liner, explosive, case, and other structures. The reactive composite jet formation characteristics are significantly affected by the explosive type, the cone angle of the liner, and the liner thickness. A simulation of reactive composite jets’ formation driven by different explosives was carried out to show how the explosive type affects the reactive composite jet formation characteristics, including TNT, Comp B, PBX, and 8701. The material parameters for TNT, Comp B, and PBX were derived from the material library in AUTODYN. The interior temperature distribution of jets with different explosive types at 1.0 CD standoff is shown in Figure 9. As shown in Figure 9, the divergence and expansion degree of the reactive composite jet tip becomes more obvious with increasing explosive detonation pressure, indicating that higher explosive detonation pressure is not conducive to the morphological stability of the reactive composite jet. When the explosive detonation pressure decreases continuously, the disaggregation problem of the reactive composite jet is improved, and the divergence degree of the jet tip is reduced.

Figure 9 also shows that the temperature of most elements from the jet tip to the junction of the jet and slug is higher than 800 K when the explosive is 8701. When the charge is Comp B or PBX, which has a lower detonation pressure, elements with temperatures that exceed 800 K partially decrease compared with that of 8701. The jet driven by TNT charge attains the maximum temperature at the axis of its tip without reaching 800 K. TNT is a relatively ideal explosive considering only the jet shape and temperature distribution of the reactive composite jet. It should be noted that the tip velocity, condensability, and temperature distribution of the reactive composite jet share a comparably significant influence on the penetration effect in practical engineering applications. Certain application requirements should be considered when choosing an explosive.

Figure 10 presents the velocity distribution of the reactive composite jet with different explosive types. As shown in Figure 10, the tip velocity of the reactive composite jet increases with the increase in the explosive detonation pressure. The tip velocity of the reactive composite jet driven by 8701 reaches a maximum value of 8330 m/s while the velocity of the reactive composite jet tip driven by TNT only reaches 6370 m/s, which is much lower than the others.

### 3.2. Cone Angle of the Reactive Liner

The cone angle of the reactive liner is one of factors that has a significant influence on the jet formation characteristics. Studies have indicated that a small cone angle leads to a high jet tip velocity while the effective mass of the jet decreases. Conversely, by increasing the cone angle, the jet tip velocity decreases while the effective mass of the jet increases. Simulation of reactive composite jets’ formation at different cone angles was carried out to show how the cone angle affects the reactive composite jet formation characteristics, including 45°, 50°, 55°, and 60°. The interior temperature distribution of the reactive composite jet at different cone angles at 1.0 CD standoff is shown in Figure 11. Figure shows a decrease in the temperature gradient of the reactive composite jet as the cone angle increases. More specifically, when the reactive liner cone angle is 45°, the high-temperature area of the reactive composite jet is most widely distributed, which almost covers the entire jet tip and the junction of the jet and slug. When the cone angle gradually increases to 60°, the high-temperature area is found at the axis of the tip and the junction of the jet and slug.

Figure 12 presents the velocity distribution of the reactive composite jet influenced by the cone angle, indicating a contradictory change between the jet tip velocity and cone angle. The jet tip velocity reaches a maximum value of 8750 m/s and a minimum value of 7600 m/s when the cone angles are 45° and 60°, respectively. In addition, the cone angle of the liner has a significant effect on the shape of the reactive composite jet. Under the given conditions of the shaped charge structure and explosive type, the tip of the reactive composite jet is more dispersive when the cone angle is small while an increasing liner cone angle will lead to a decrease in both the expansion effect of the jet tip and the slug volume, and an increase in the effective jet mass.

The main reason is that the increase in the cone angle can reduce the velocity gradient of the jet, which improves the cohesion and continuity of the jet. However, an overlarge cone angle will affect the penetration performance of the reactive composite jet by reducing the jet tip velocity. It follows that a suitable cone angle is a crucial factor for guaranteeing the lethality of the reactive composite jet when designing the structure of the shaped charge with the reactive liner.

### 3.3. Wall Thickness of the Reactive Liner

The optimum thickness of the liner is closely related to the density of the liner material, the cone angle, the charge diameter, and the case. Simulation of reactive composite jets’ formation at different liner thicknesses was carried out to show how the liner thickness influences the reactive composite jet formation characteristics, including 0.08, 0.10, and 0.12 CD, while the other factors remained the same.

The temperature distribution of the reactive composite jet with different liner thicknesses at 1.0 CD standoffs is shown in Figure 13. There is a gradual decrease in the high-temperature region inside the reactive composite jet as the liner thickness increases. The whole jet tip and the junction region between the jet tip and slug is almost in the high-temperature area when the liner thickness is 0.08 CD while the high-temperature area covers half of the area of the jet tip and the junction between the jet and slug when the liner thickness is 0.12 CD. Note that the reactive composite jet formation by different liner thicknesses shares a similar divergence at the jet tip, although the reactive composite jet still does not agglomerate.

Figure 14 shows the velocity distribution of the jet tip influenced by the liner thickness. The velocity of the jet tip gradually decreases with increasing liner thickness. When the thickness is 0.08, 0.10, and 0.12 CD, the jet tip velocity reaches a maximum value of 9025, 8330, and 8000 m/s, respectively. It can also be seen that by increasing the liner thickness, the velocity of the reactive composite jet tip decreases slower when the standoff increases.

## 4. Reaction Characteristic of the Reactive Composite Jet

### 4.1. Chemical Reaction of PTFE/Al Reactive Composites

PTFE is a complex semi-crystalline material with the chemical formula—(CF2-CF2)—which has great stability and chemical inertness at room temperature. To better understand the thermal decomposition and reaction process of PTFE/Al reactive composites, the chemical thermal reaction behavior of 73.5 wt.% PTFE/26.5 wt.% Al mixture and pure PTFE was investigated using DSC/TG experiments. The experiment was carried out from room temperature to 1120 K (850 °C) in an argon atmosphere, and the heating rate was 20 K/min. In the reaction process of PTFE/Al reactive composites, PTFE first decomposes to release C_2_F_4_. With the passage of time, the Al particles in the powder react with the fluoropolymer, releasing many reaction heat and gas products. In an anoxic system, the main reactions are as follows:-(C_2_F_4_)n- → n C_2_F_4_ (g)(3)
4Al + 3C_2_F_4_ → 4AlF_3_ (g) + 6C(4)
4Al + C_2_F_4_ → 4AlF (g) + 2C(5)
2Al + C_2_F_4_ → 2AlF_2_ (g) + 2C(6)

The results of the DSC/TG experiments for reactive composites are displayed in Figure 15. It shows that an endotherm, such as in melting endothermic, appears as a peak while an exotherm, such as in reaction exotherm, appears as a valley.

The DSC and TG thermal analysis results show that reactive composites maintain good chemical stability below 600 K. Comparing Figure 15a,b, it can be found that the matrix material PTFE begins to melt and absorb heat at 600 K, with an endothermic peak of 616.7 K. The PTFE begins to crack and release heat after reaching a molten state, but at this time, the amount of the cracking matrix material (PTFE) is still very small, and the weight loss per hour is only about 2%, and it does not react with the metal fuel (Al). The TG curve suggests that the sample weight dropped sharply from 783 K. It reaches the second endothermic peak at 873 K and a large amount of C_2_F_4_^+^ ions are produced at this time. As the temperature continues to rise, due to the increase in the ion concentration and temperature, the reaction between the Al and PTFE decomposition products accelerates, generating a large amount of heat and forming an exothermic valley. Comparing Figure 15a,b, when the temperature reaches 963 K, the unreacted Al melts and a third endothermic peak appears. The endothermic peak and the exothermic valley after 813 K are the result of the superposition of the heat absorbed by the polymer cracking and melting of the material and the heat released by the reaction of the reactive composites.

### 4.2. X-ray Experimental Setup

The PTFE/Al reactive composites have metal-like strength and dual properties of explosive energy, so they exhibit unique mechanical and chemical coupling response behavior under the detonation of the shaped charge. Therefore, it is necessary to verify whether the PTFE/Al liner can form an effective jet. In this section, pulse X-ray testing was used to demonstrate the formation of the reactive composite jets and to study the response of reactive composites. Using the method of combining the pulse X-ray experiment and static explosion experiment, the formation process and dynamic response of the reactive composite jet were studied.

The preparation process of the reactive liner used in the experiment mainly consists of 3 steps: at first, the reactive material powder with 73.5 wt.% PTFE and 26.5 wt.% Al was preliminarily mixed. The average size of the PTFE and Al particles was 100 and 44 μm, respectively. The preliminarily mixed powder was mixed in the planetary centrifugal mill for 3 h and the fully mixed powder was placed in an 82 °C vacuum drying oven for about 24 h. Then, 49.1 g of reactive material powder were weighed and put into the mold. The mold was pressed for 30 s at a pressure of 300 MPa and placed to eliminate residual stress. Finally, the pressed reactive liner was placed in a nitrogen-filled sintering furnace for sintering. Specifically, the sample was heated at a rate of 50 °C/h in the sintering furnace, and when the temperature was increased to a maximum temperature of 380 °C, the sample was sintered at 380 °C for 6 h. Then, the temperature was cooled at a rate of 50 °C/h. When cooled to a maximum temperature of 310 °C, it was sintered at 310 °C for 4 h, and then cooled down to room temperature with the furnace [16]. The reactive liner is shown in Figure 16.

The microstructure features of the sintered sample were characterized by a Regulus-8230 SEM (scanning electron microscope), and the results are shown in Figure 17. The sintered sample has a dense structure and a clear microscopic structure, and the Al particles are nearly circular and relatively uniformly distributed in the PTFE matrix.

The reactive shaped charge used in the experiment consists of a reactive liner, a main charge 8701, a case, and a detonating device. When the main charge is detonated, the detonation wave propagates in the main charge along the axial direction, and then the detonation wave acts on the outer wall of the reactive liner, and the liner collapses toward the axis to form the reactive composite jet and a slug. As time progresses, the penetrator will continue to elongate and become a longer reactive composite jet at a velocity generally between 7000 and 8500 m/s. The experimental principle and site layout of the reactive composite jet formation pulse X-ray are shown in Figure 18.

### 4.3. X-ray Experimental Results

The typical reactive composite jet formation behavior is shown in Figure 19. Taking the detonation time of the main charge as 0 μs, pulse X-rays were taken at 2 moments of 15 and 21 μs. The jet tip shows different degrees of divergence and expansion effects at two moments. Figure 19a shows that the tip of the PTFE/Al reactive composite jet reaches approximately the position of 1.0 CD standoff, and the corresponding tip velocity of the reactive composite jet is approximately 8200 m/s. At this moment, under the detonation driving action of the main charge, a reactive composite jet with an excellent performance can be formed by the reactive liner. The reactive composite jet tip and slug shape are relatively clear, and the jet continuity and coaxiality are also good. Moreover, the slug is larger, the shape is clearer, and the imaging color is brighter. This indicates that most of the mass of the reactive liner forms part of the slug, leading to higher density and compactness. Compared with the slug, the reactive composite jet has a lighter color, which is caused by the expansion of the jet tip during the formation process, which is consistent with the calculation results of the numerical simulation. It should be noted that the higher the density of the reactive composite jet, the brighter the color and the clearer the contour in the X-ray image.

For Figure 19b, the position of the reactive composite jet tip reached is approximately 2.0 CD standoff, and the jet tip velocity is approximately 7810 m/s. At this moment, the reactive composite jet still has excellent continuity, symmetry, and coaxiality. However, the jet profile is unclear, especially the jet tip, and it is difficult to distinguish the contour. Compared with the radiograph at the 1.0 CD standoff, the jet at the 2.0 CD standoff has obvious expansion, resulting in reduced density, a blurred jet profile, and lighter color in the X-ray image. At the junction of the jet and slug, the expansion and divergence effect of the reactive composite jet can be clearly seen. In addition, the radiograph of 21 μs shows that the reactive composites of the jet tip experienced the local chemical reaction at this moment. However, owing to the relatively small mass of the jet tip, its deflagration reaction has not caused the violent chemical reaction of the jet tail and slug section. This is consistent with the expansion effect and high temperature distribution of the reactive composite jet in the numerical simulation.

When the steel plate was placed at a height of 10 CD standoff (see in Figure 18), the experimental photograph of the reactive composite jet penetrating steel plate is shown in Figure 20. The experimental result shows that the penetration depth is approximately 8 mm and the penetration hole diameter is 8 mm. In addition, the surface of the steel target is almost completely covered by the oxidation reaction product C, which corresponds to the analysis result in Section 3.1. As such, it can be inferred that although the tip of the reactive composite jet has reacted at the 2.0 CD standoff, the chemical reaction of the jet tip will not immediately cause the complete and violent deflagration reaction. This is because once the reactive composite jet experiences a violent deflagration reaction, it has no penetration ability and cannot produce an armor penetration effect.

According to the analysis with numerical simulation results, driven by the detonation of the main charge, while the reactive liner forms the jet, the internal temperature of the reactive composite jet will increase due to the loading and unloading of the shock wave and the effect of heat conduction. Moreover, the temperature will also rise due to the plastic deformation during the liner collapse process and the jet stretching process [25]. However, unlike the self-sustaining reaction of traditional energetic materials, the chemical action of the reactive composites is not self-sustaining owing to the PTFE/Al materials with higher strength [23]. After the detonation of the main charge, the formed reactive composite jet still has a good kinetic energy penetration ability in the delay time.

In addition, although the peak pressure of the detonation shock wave can reach about 20 GPa, the activation pressure and loading environment are different within the reactive liner element due to the reactive liner with having a large axial space size, which means the formed reactive composite jet cannot be activated at the same time. In other words, the chemical reaction time of the reactive composite jet varies significantly along its axis. According to Figure 6, the tip of the reactive composite jet is generally at a high temperature, and the reactive composites of the jet tip will first experience PTFE decomposition and oxidization reaction, which fits well with the X-ray experimental results. Therefore, the chemical reaction of the reactive composite jet has a delay characteristic. Generally, the period from the decomposition of the reactive composites to the occurrence of a violent deflagration reaction is called the reaction delay time. In this period of time, the reactive composite jet with the higher velocity will continue to be stretched and move forward until it impacts the steel plate. The high-speed interaction between the reactive composite jet and the steel plate will activate the reactive composites again, resulting in the residual jet and slug completely undergoing a violent deflagration reaction, which causes the penetration process of the reactive composite jet to terminate.

## 5. Conclusions

In this study, the density, velocity, and temperature distribution of the reactive composite jet during the formation process were studied, and the influence of the shaped charge conditions on the jet formation behavior was also analyzed. Several conclusions are presented as follows:(a)Compared with the traditional Cu jet, numerical simulations showed that the reactive composite jet has an obvious expansion effect, accompanied by a significant decrease in the overall density. When the reactive composite jet tip reaches the 2.0 CD standoff, the density of the reactive composite jet element drops to 2.0 g/cm^3^, especially the density of the jet tip element, which is only approximately 1.6 g/cm^3^.(b)From the temperature distribution of the reactive composite jet, the reactive composite jet tip and its axis present high-temperature zones while the temperature of the slug and the two wings is lower. For a given caliber RLSC, the type of explosive, cone angle, and wall thickness of the reactive liner have significant effects on the temperature distribution. As the detonation pressure of the explosive increases, the proportion of the high-temperature zone in the reactive composite jet tip will also increase. However, by increasing the cone angle and wall thickness of the reactive liner, the proportion of the high-temperature zone in the reactive composite jet tip will decrease.(c)A DSC/TG and pulse X-ray experiment was carried out to analyze the reaction characteristic and formation characteristics of the reactive composite jet. Combining the temperature distribution of the reactive composite jet given by the numerical simulation and the experimental results of DSC/TG, it is concluded that the distribution of the high-temperature zone of the reactive composite jet is mainly concentrated at the tip and axis of the jet, and the reactive composite jet tip will react first. The X-ray experiment results show that the reactive composite jet has excellent coaxiality, continuity, and symmetry within a certain range of standoff. The formation morphology of the reactive composite jet is greatly affected by the reaction delay time. As the time increases, the contour of the reactive composite jet is gradually blurred, the tip expansion phenomenon is intensified, and the density decreases. However, due to the non-self-sustaining reaction characteristics of the reactive composites, the reactive composite jet still has the ability of kinetic energy penetration within a reaction delay time.(d)In this study, the distribution of the density, temperature, and velocity field in the PTFE/Al reactive composite jet formation was only analyzed in the form of a distribution diagram and observation points, and the state of all SPH particles cannot be monitored yet. In future work, we will carry out the secondary development of AUTODYN finite element analysis software. Based on the SPH algorithm, the state of any particle in the process of reactive composite jet formation will be tracked, and high-temperature particles will be screened out. Therefore, the PTFE/Al composite material in the high-temperature region can be replaced to realize the control of the reaction time of the reactive composite jet.

## Figures and Tables

**Figure 1 materials-15-01268-f001:**
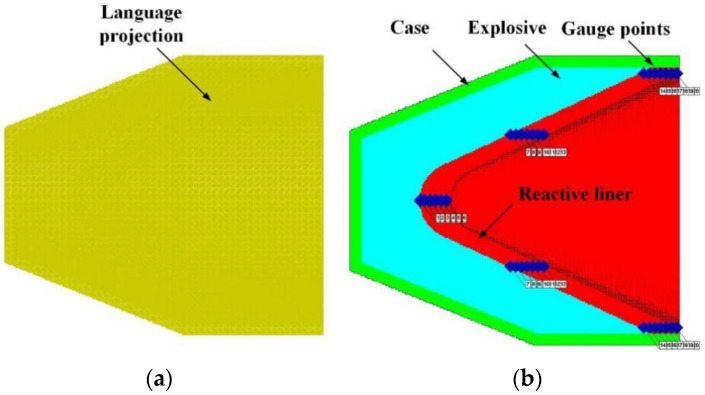
Numerical model of RLSC: (**a**) simulation meshing and (**b**) SPH particle filling.

**Figure 2 materials-15-01268-f002:**
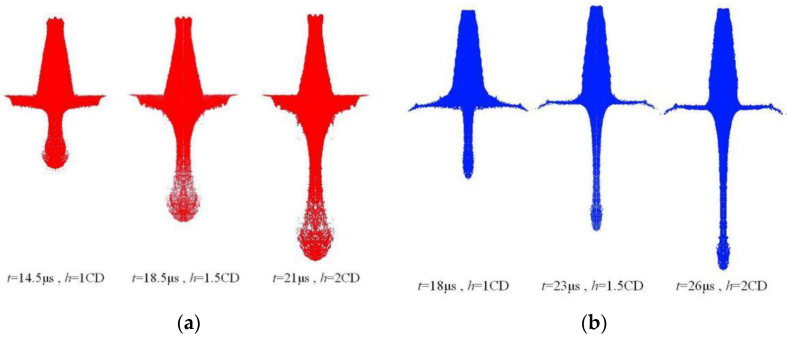
Comparison of jet formation characteristics: (**a**) reactive composite jet and (**b**) Cu jet.

**Figure 3 materials-15-01268-f003:**
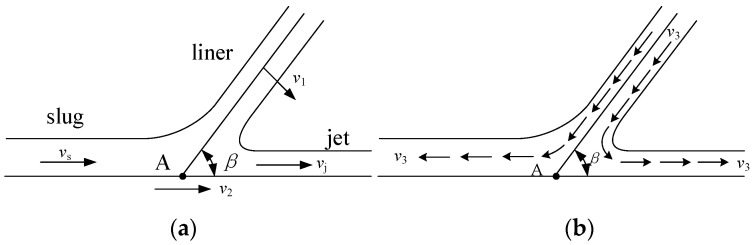
Reactive composite jet formation process (**a**) static coordinate system (**b**) and dynamic coordinate system.

**Figure 4 materials-15-01268-f004:**
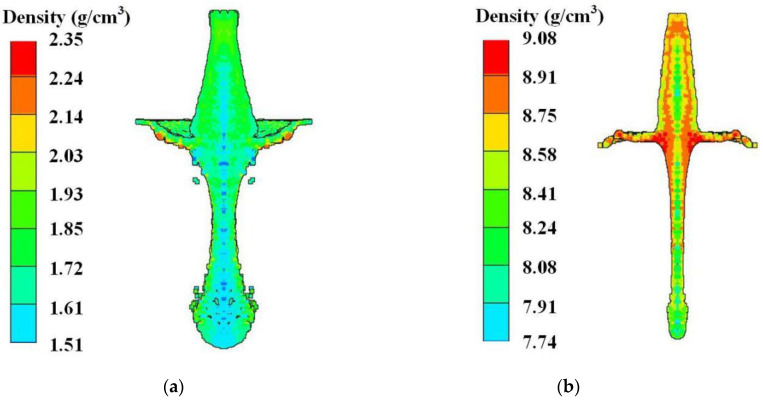
The jet density distribution at 2.0 CD standoff: (**a**) Reactive composite jet and (**b**) Cu jet.

**Figure 5 materials-15-01268-f005:**
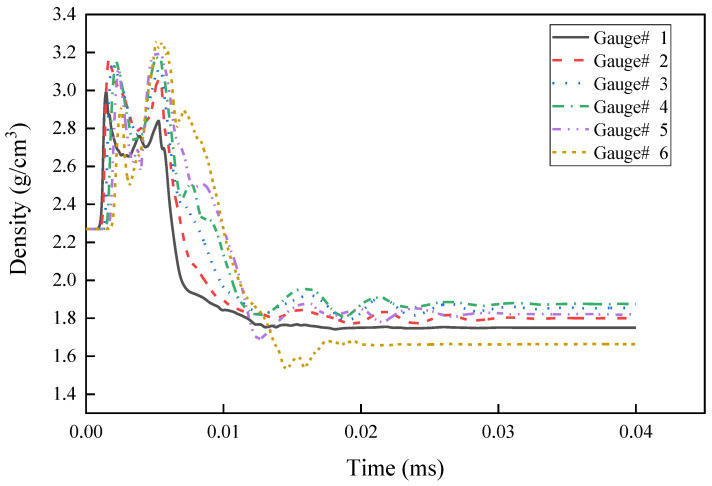
Density of the axis of the reactive composite jet with time.

**Figure 6 materials-15-01268-f006:**
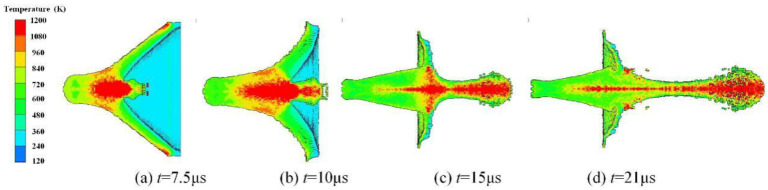
Typical temperature changes during reactive composite jet formation.

**Figure 7 materials-15-01268-f007:**
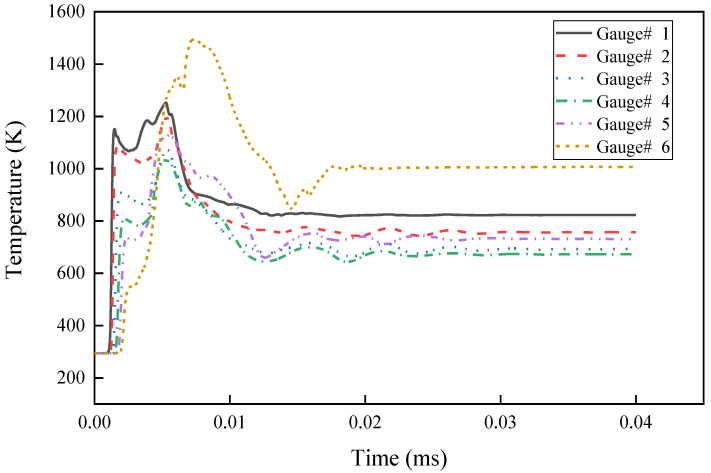
Temperature of the axis of the composite jet with time.

**Figure 8 materials-15-01268-f008:**
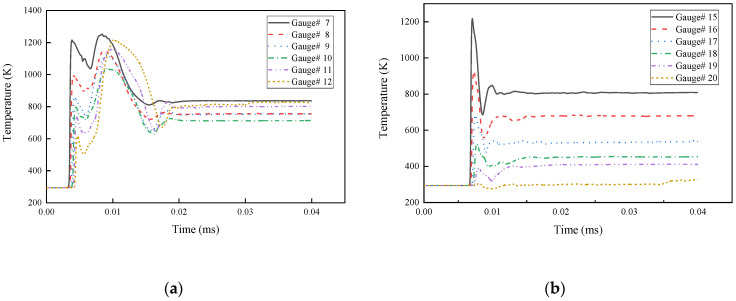
Temperature of the reactive composite jet change with time: (**a**) middle of the reactive liner (**b**) and bottom of the reactive liner.

**Figure 9 materials-15-01268-f009:**
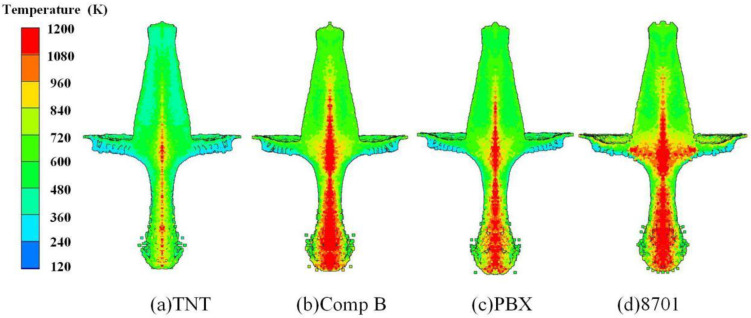
Temperature distribution of the reactive composite jet influenced by explosives.

**Figure 10 materials-15-01268-f010:**
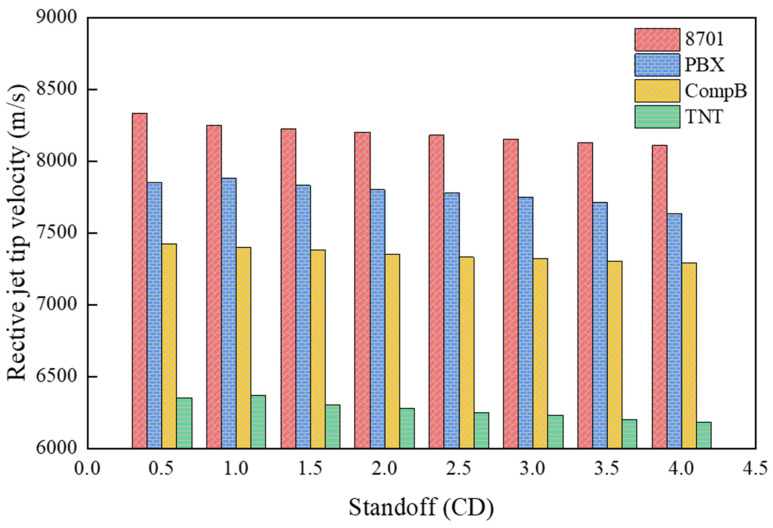
The tip velocity of the reactive composite jet influenced by explosives.

**Figure 11 materials-15-01268-f011:**
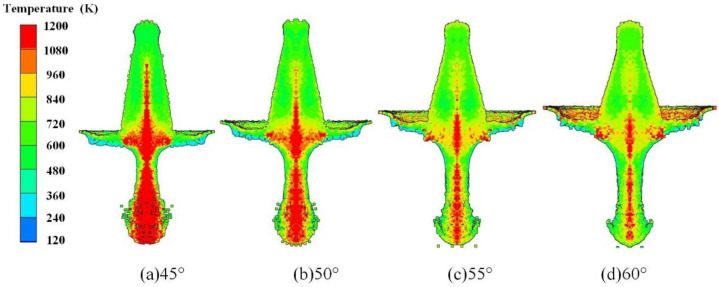
Reactive composite jet temperature gradient distribution influenced by the cone angle.

**Figure 12 materials-15-01268-f012:**
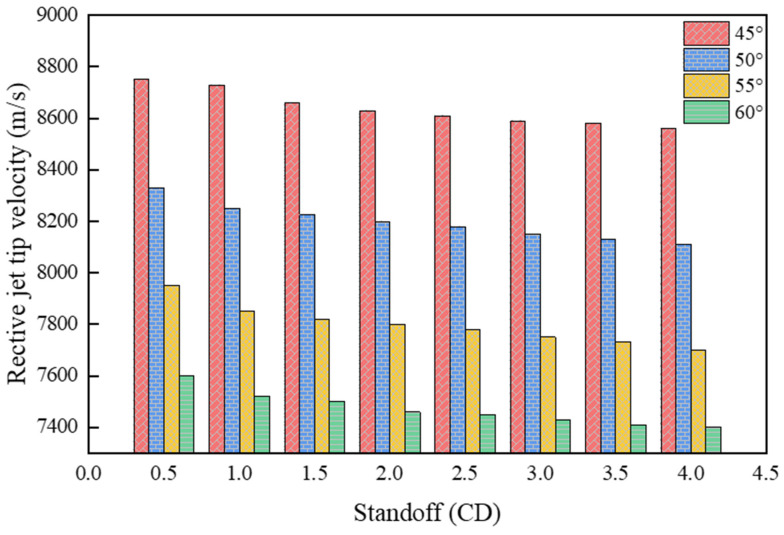
Reactive composite jet tip velocity influenced by the cone angle.

**Figure 13 materials-15-01268-f013:**
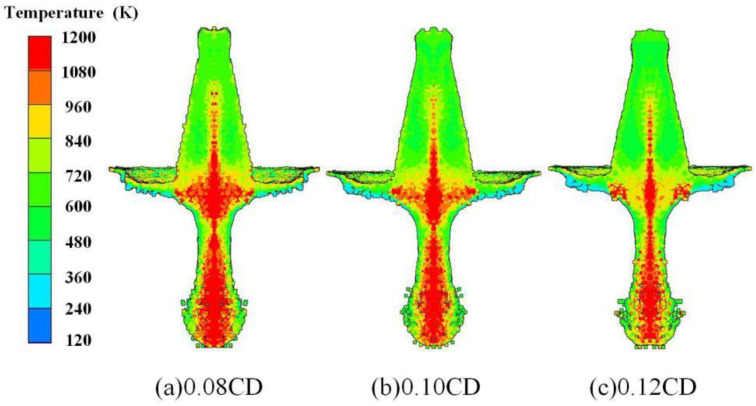
Reactive composite jet temperature distribution influenced by the wall thickness.

**Figure 14 materials-15-01268-f014:**
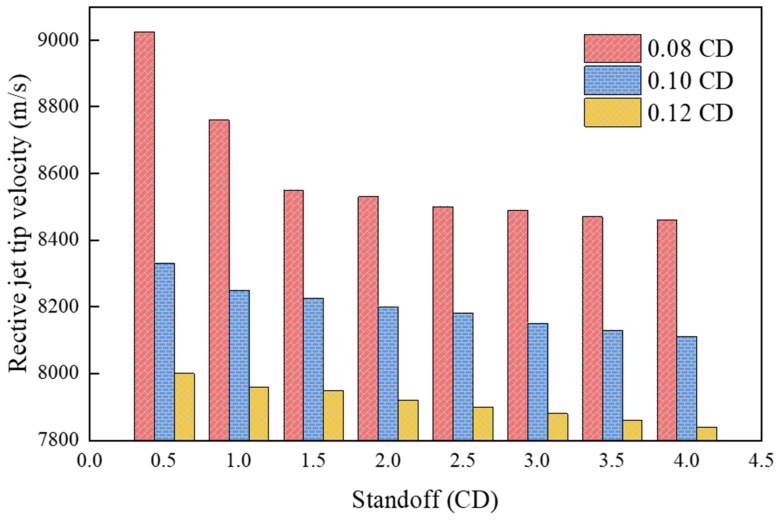
Reactive composite jet tip velocity influenced by the liner thickness.

**Figure 15 materials-15-01268-f015:**
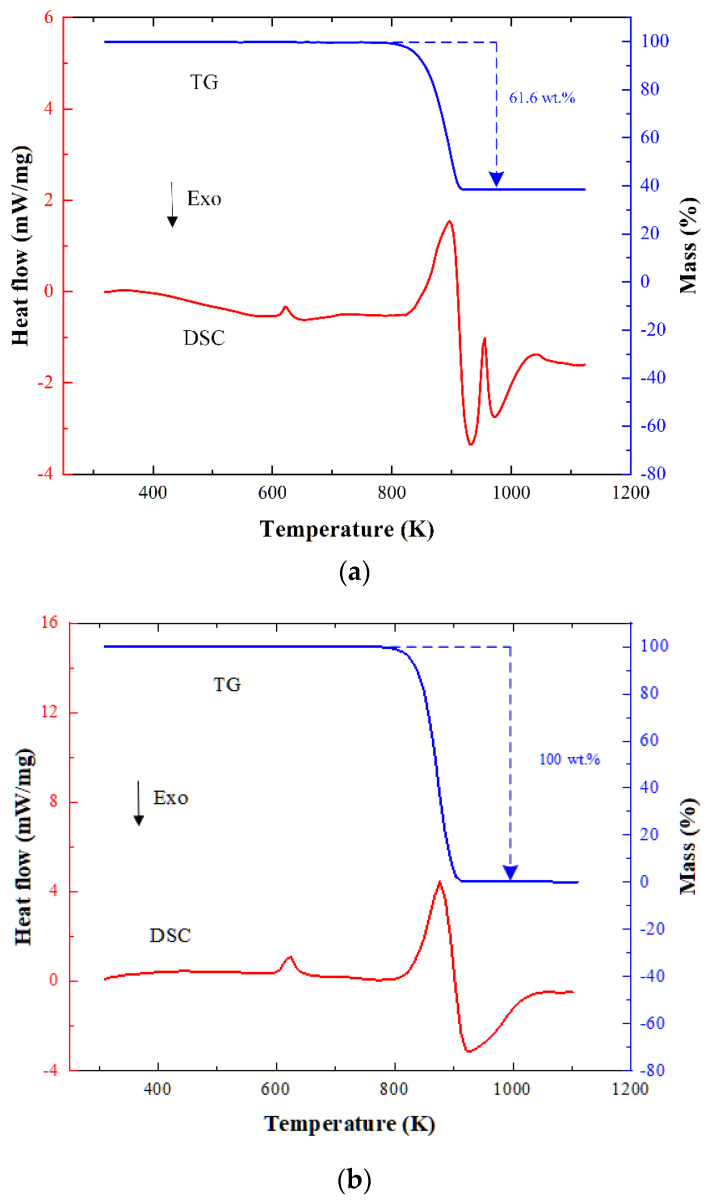
DSC/TG results of (**a**) PTFE/Al reactive composites and (**b**) pure PTFE.

**Figure 16 materials-15-01268-f016:**
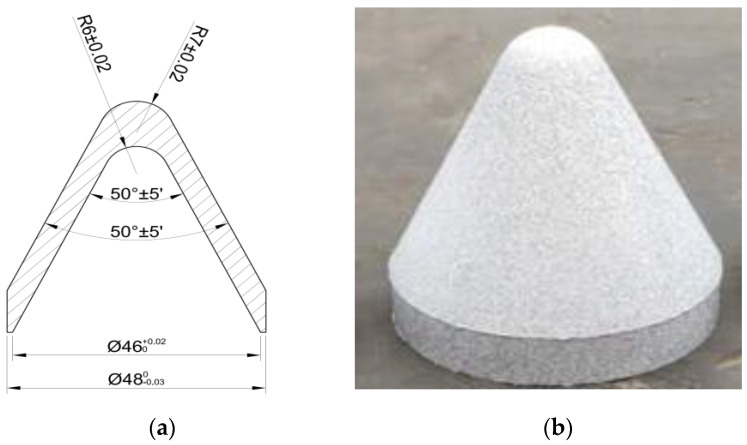
Structure and sample of the reactive liner: (**a**) structure and (**b**) sample.

**Figure 17 materials-15-01268-f017:**
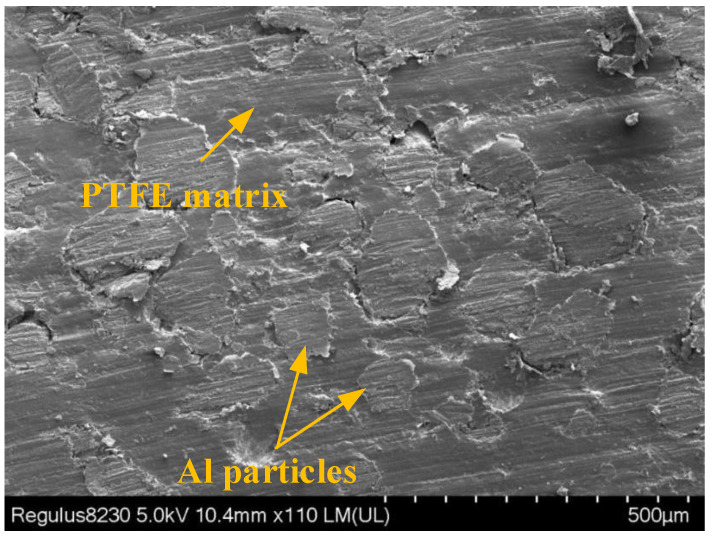
SEM of Al/PTFE reactive composites.

**Figure 18 materials-15-01268-f018:**
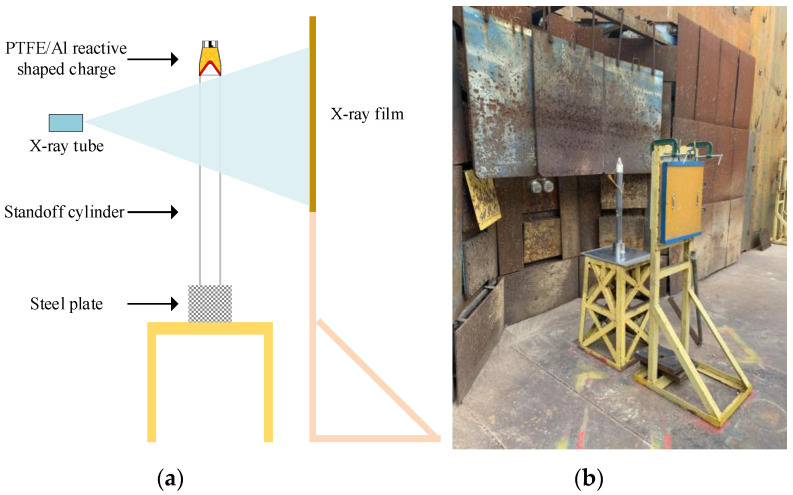
Pulse X-ray experiment (**a**) schematic and (**b**) setup.

**Figure 19 materials-15-01268-f019:**
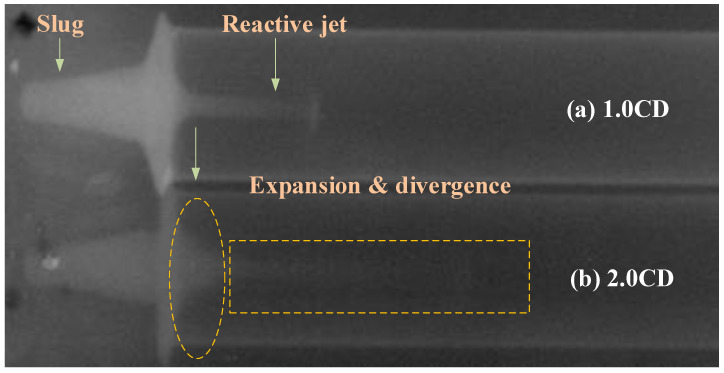
X-ray image of reactive composite jet formation (**a**) *t* = 15 μs and (**b**) *t* = 21 μs.

**Figure 20 materials-15-01268-f020:**
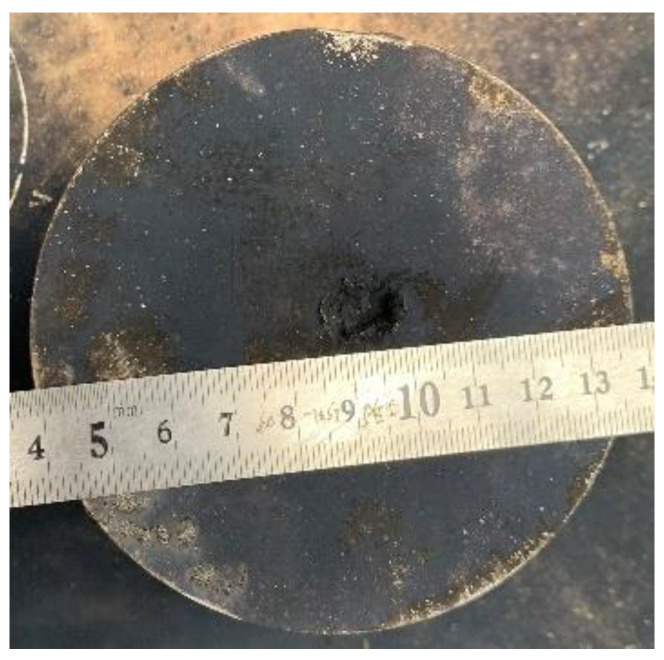
Experimental result of the steel plate impacted by the PTFE/Al reactive composite jet.

**Table 1 materials-15-01268-t001:** Material strength models and EOSs of RLSC.

Part	Materials	EOS	Strength	Erosion
Liner	PTFE/Al	Shock	Johnson Cook	None
Explosive	8701	JWL	None	None
Case	#45 steel	Shock	Johnson Cook	None

**Table 2 materials-15-01268-t002:** Johnson–Cook strength model parameters of the reactive liner and #45 steel.

Material	*ρ* (g/cm^3^)	*G* (Gpa)	*A* (Mpa)	*B* (Mpa)	*n*	*C*	*m*	*T_m_* (K)	*T_room_* (K)
Reactive liner	2.27	0.67	8.04	250.6	1.8	0.4	1	500	294
#45 steel	7.83	77	792	510	0.26	0.014	1.03	1793	300

**Table 3 materials-15-01268-t003:** JWL EOS parameters of the 8701 explosive.

*ρ* (g/cm^3^)	*D* (km/s)	*P*_CJ_ (GPa)	*e* (GPa)	*A* (GPa)	*B* (GPa)	*R* _1_	*R* _2_	*ω*	*v* _0_
1.71	8.315	28.6	8.499	524.23	7.678	4.2	1.1	0.34	1.00

## Data Availability

Not applicable.

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
