# Peer review of "Formation Behavior and Reaction Characteristic of a PTFE/Al Reactive Jet"

_materials, 2022, doi:10.3390/ma15031268_

Round 1
Reviewer 1 Report
Manuscript in need of revision

Author Response
Response to the comments from reviewer 1
Dear reviewer:
First, we would like to thank the reviewer and the Editorial Office for their feedback on our paper. Your work on reviewing this manuscript is well appreciated. The feedback and comments have given very valuable and helpful advice for improving the article. We have made careful revisions in the manuscript according to the comments and replied the point by point. The changes made in the revised manuscript have been highlighted by using yellow background. We hope the revised manuscript can meet the journal’s desired standard. We look forward to learning your response to our submission.
Yours sincerely,
Point 1: The subject matter of the journal refers to the properties of materials, while the article describes how the jet is formed. The manuscript needs to be strengthened by the properties of the materials and their mixtures.
Response 1: Thanks a lot for your valuable comments. We have made changes to the manuscript, first, to highlight the properties of PTFE/Al composite materials, we supplement the research progress of PTFE/Al materials and their mixtures in the Introduction, references [3-9] are newly added. The specific modifications are as follows:
Fluoropolymer-based reactive composites are solid energetic materials mixed with reactive metal powder, alloy powder or intermetallic compound in polymer powder (typically PTFE), which can release chemical energy under high dynamic load and high strain rate conditions. Due to its unique properties, reactive composites have high application value in military and civilian fields [1, 2]. The PTFE/Al composite material is a typical reactive material, which is prepared by uniformly filling Al particles into a PTFE matrix, and cold-pressing and sintering. This material has become a benchmark for studying the properties and applications of reactive materials. Recently, for PTFE/Al composites, the formulations and fabrications [3], impact initiation [4] and chemical reaction [5] have been studied. Especially in terms of formulation, by adding high-density W powder to the PTFE/Al composites [6], the strength can be improved; by adding metal powders or metal oxides, such as MoO3 [7], Bi2O3 [8], CuO [9], etc., to the PTFE/Al composites, the heat release and gas product amount can be improved.
[3] C Ge, W Maimaitituersun, Y X Dong, et al. A study on the mechanical properties and impact-induced initiation characteristics of brittle PTFE/Al/W reactive materials [J], Materials 2017 (10) (2017) 452.
[4] F Bin, L Yuchun, W Shuangzhang, et al. A crack-induced initiation mechanism of Al-PTFE under quasi-static compression and the influencing factors [J], Mater. Des. 108 (15) (2016) 411–417.
[5] Z Song, L Jinxu, Y. Min et al. Effects of multi-component co-addition on reaction characteristics and impact damage properties of reactive materials [J], Mater. Des. 153 (5) (2018) 1–8.
[6] T Sun, Y F Zheng, Y Yuan, et al. Impact-Initiation Sensitivity of High-Temperature PTFE-Al-W Reactive Materials. Crystals. 2022, 12(1), 30.
[7] J Y Huang, X Fang, S Z Wu, et al. Mechanical Response and Shear-Induced Initiation Properties of PTFE/Al/MoO3 Reactive Composites. Materials. 2018, 11, 1200.
[8] Y Yuan, B Q Geng, T Sun, et al. Impact-Induced Reaction Characteristic and the Enhanced Sensitivity of PTFE/Al/Bi2O3 Composites. Polymers. 2019, 11(12), 2049.
[9] L L Ding, J Y Zhou, W H Tang, et al. Impact Energy Release Characteristics of PTFE/Al/CuO Reactive Materials Measured by a New Energy Release Testing Device. Polymers 2019, 11, 149.
Subsequently, in Section 4.1 of the manuscript, we made some supplements to the properties of PTFE materials, and carried out DSC/TG experiments on the matrix PTFE material, which was convenient for comparative analysis with PTFE/Al composite materials. The specific modifications are as follows:
PTFE is a complex semi-crystalline material with the chemical formula - (CF2-CF2) - which has great stability and chemical inertness at room temperature. To better understand the thermal decomposition and reaction process of PTFE/Al reactive composites, the chemical thermal reaction behavior of 73.5 wt.% PTFE/26.5 wt.% Al mixture and pure PTFE are researched by DSC/TG experiments.
The results of the DSC/TG experiments for PTFE/Al reactive composites and pure PTFE are displayed in Fig. 15. It shows that an endotherm, such as in melting endothermic, appears as a peak while an exotherm, such as in reaction exotherm, will appear as a valley.
(a)
(b)
Fig. 15. DSC/TG results of (a) PTFE/Al reactive composites and (b) pure PTFE.
DSC and TG thermal analysis results show that reactive composites maintain good chemical stability below 600 K. Comparing Fig.15 (a) and (b), it can be found that the matrix material PTFE begins to melt and absorb heat at 600 K, with an endothermic peak of 616.7 K. The PTFE begins to crack and release heat after reaching a molten state, but the amount of the cracking matrix material (PTFE) is still very small, and the weight loss per hour is only about 2%, it does not react with the metal fuel (Al). The TG curve suggests that the sample weight dropped sharply from 783 K. It reaches the second endothermic peak at 873 K and a large amount of C2F4+ ions are produced at this time. As the temperature continues to rise, due to the increase in ion concentration and temperature, the reaction between Al and PTFE decomposition products accelerates, generating a large amount of heat and forming an exothermic valley. Comparing Fig.15 (a) and (b), when the temperature reaches 963 K, the unreacted Al melts and a third endothermic peak appears. The endothermic peak and the exothermic valley after 813 K are the result of the superposition of the heat absorbed by the polymer cracking and melting of the material and the heat released by the reaction of the reactive composites.
Finally, we carried out SEM (scanning electron microscope) experiments on the prepared reactive composite liner in Section 4.2 of the manuscript to be more in line with the journal theme. The specific modifications are as follows:
Finally, the pressed reactive liner is placed in a nitrogen-filled sintering furnace for sintering. Specifically, the sample was heated at a rate of 50°C/h in the sintering furnace, and when the temperature was increased to a maximum temperature of 380°C, the sample was sintered at 380°C for 6 hours; then the temperature was cooled at a rate of 50°C/h, When cooling to a maximum temperature of 310°C, sinter at 310°C for 4 hours, and then cool down to room temperature with the furnace. The reactive liner is shown in Fig. 16.
The microstructure features of the sintered sample was characterized by a Regulus-8230 SEM (scanning electron microscope), and the results are shown in Fig. 17. The sintered sample has a dense structure and a clear microscopic structure, and the Al particles are nearly circular and relatively uniformly distributed in the PTFE matrix.
Fig. 17. SEM of Al/PTFE reactive composites.
Point 2: Long abstract. Manuscript in need of revision.
Response 2: Thanks for your reminding. We have made certain deletions and revisions to the abstract of the manuscript to make it more self-explanatory, and at the same time highlight the research focus and innovation of the manuscript. Specific modifications to the abstract are as follows.
Abstract:The temperature and density distribution of aluminum particle filled polytetrafluoroethylene (PTFE/Al) reactive composite jet were investigated by combining numerical simulation and experimental study. Based on the platform of AUTODYN-3D code,the Smoothed Particle Hydrodynamics (SPH) algorithm was used to study the evolution behaviors and distribution regularity of morphology, density, temperature, and velocity field during the formation process of reactive composite jet. The reaction characteristic in forming process was revealed by combining the distribution of high temperature zone in numerical simulation and the Differential Scanning Calorimeter/Thermo-Gravimetry (DSC/TG) experiment results. The results show that the distribution of the high temperature zone of the reactive composite jet is mainly concentrated in the jet tip and the axial direction, and the reactive composite jet tip reacts first. Combined with density distribution in the numerical simulation and pulsed X-ray experimental result, the forming behavior of reactive composite jet was analyzed. The results show that the reactive composite jet has an obvious expansion effect, accompanying with a significant decrease in the overall density.
Point 3: Currently, there are many programs that allow you to see the distribution of temperatures. What is the novelty of your research from the article is not obvious.
Response 3: Thanks a lot for your valuable comments. It needs to be explained to the reviewer that although there are many programs that can calculate the temperature distribution of materials, there are still few applications in jet forming of shaped charges. In the previous studies (as references [4, 6] in the original manuscript), the Euler algorithm was almost used for the forming of the jet, but the SPH algorithm selected in this manuscript can better simulate and explain the expansion of the jet tip during the forming process of the PTFE/Al reactive composite jet and divergent phenomena, and the jet morphology is closer to X-ray experiment. Moreover, in the previous research (as references [12, 13] in the original manuscript), the research focused more on the damage effect of the reactive jet, while ignoring the reaction characteristic of the reactive jet. In this paper, the reaction characteristic of the PTFE/Al reactive composite jet is analyzed through the distribution of the temperature field. Therefore, this manuscript is innovative to a certain extent.
It is necessary to explain to the reviewers that this manuscript is a phased achievement. At present, we are carrying out the secondary development of AUTODYN finite element analysis software. Based on SPH algorithm, the state of any particle in the process of reactive composite jet forming will be tracked, and high-temperature particles will be screened out. Thereby, the composite material in the high temperature region can be replaced to realize the control of the reaction time of the reactive composite jet.
Therefore, we have added a new paragraph to the conclusion to outline the challenges in the current research, future work, and recommendations. The specific modifications are as follows:
(d) In this paper, the distribution of density, temperature and velocity field in PTFE/Al reactive composite jet forming is only analyzed in the form of distribution diagram and observation points, and the state of all SPH particles cannot be monitored yet. In the future work, we will carry out the secondary development of AUTODYN finite element analysis software. Based on SPH algorithm, the state of any particle in the process of reactive composite jet forming will be tracked, and high-temperature particles will be screened out. Therefore, the PTFE/Al composite material in the high temperature region can be replaced to realize the control of the reaction time of the reactive composite jet.
Point 4: The purpose of the study is not obvious, what is this type of jet needed to destroy?
Response 4: Thanks a lot for your reminding and comments. We are very sorry to have confused you. Reactive composite jet and its shaped charge warhead technology have attracted much attention in recent years, and are currently one of the hot frontier research directions in the field of efficient damage. It can be used to damage objects such as armor, concrete and ships. Under the explosive driving action of the shaped charge, the reactive composite jet formed by the reactive material liner, not only can it penetrate the target like a traditional metal jet, but more importantly, the reactive composite jet can activate itself after penetrating the interior of the target and cause a violent explosion/deflagration reaction, releasing a large amount of chemical energy, thereby formation a more lethal killing/damaging effect on the inside of the target.
The main purpose of this paper is to study the forming behavior and reaction characteristic of the PTFE/Al reactive composite jet, so the damage target is not analyzed. We have made certain deletions and revisions to the abstract of the manuscript to make it more self-explanatory, and at the same time highlight the research focus and innovation of the manuscript. The specific modification content is in Response 2.
Thank you again for your serious and careful work. I wish you success in your work and a happy life.

Reviewer 2 Report
- The authors need to describe the novelty of the work.
- Add latest references in the introduction section.
- The introduction section needs a comparison of fluorinated compounds.
- It is strongly recommended to add a subsection, 'practical implications of this study,' outlining the challenges in the current research, future work, and recommendations, before the conclusion.
- The title should be changed.
Author Response
Response to the comments from reviewer 2
Dear reviewer:
First, we would like to thank the reviewer and the Editorial Office for their feedback on our paper. Your work on reviewing this manuscript is well appreciated. The feedback and comments have given very valuable and helpful advice for improving the article. We have made careful revisions in the manuscript according to the comments and replied the point by point. The changes made in the revised manuscript have been highlighted by using yellow background. We hope the revised manuscript can meet the journal’s desired standard. We look forward to learning your response to our submission.
Yours sincerely,
Point 1: The authors need to describe the novelty of the work.
Response 1: Thanks a lot for your valuable comments. It needs to be explained to the reviewer that although there are many programs that can calculate the temperature distribution of materials, there are still few applications in jet forming of shaped charges. In the previous studies (as references [4, 6] in the original manuscript), the Euler algorithm was almost used for the forming of the jet, but the SPH algorithm selected in this manuscript can better simulate and explain the expansion of the jet tip during the forming process of the PTFE/Al reactive composite jet and divergent phenomena, and the jet morphology is closer to X-ray experiment. Moreover, in the previous research (as references [12, 13] in the original manuscript), the research focused more on the damage effect of the reactive jet, while ignoring the reaction characteristic of the reactive jet. In this paper, the reaction characteristic of the PTFE/Al reactive composite jet is analyzed through the distribution of the temperature field. Therefore, this manuscript is innovative to a certain extent. We have made certain deletions and revisions to the abstract of the manuscript to make it more self-explanatory, and at the same time highlight the research focus and innovation of the manuscript. Specific modifications to the abstract are as follows.
Abstract:The temperature and density distribution of aluminum particle filled polytetrafluoroethylene (PTFE/Al) reactive composite jet were investigated by combining numerical simulation and experimental study. Based on the platform of AUTODYN-3D code,the Smoothed Particle Hydrodynamics (SPH) algorithm was used to study the evolution behaviors and distribution regularity of morphology, density, temperature, and velocity field during the formation process of reactive composite jet. The reaction characteristic in forming process was revealed by combining the distribution of high temperature zone in numerical simulation and the Differential Scanning Calorimeter/Thermo-Gravimetry (DSC/TG) experiment results. The results show that the distribution of the high temperature zone of the reactive composite jet is mainly concentrated in the jet tip and the axial direction, and the reactive composite jet tip reacts first. Combined with density distribution in the numerical simulation and pulsed X-ray experimental result, the forming behavior of reactive composite jet was analyzed. The results show that the reactive composite jet has an obvious expansion effect, accompanying with a significant decrease in the overall density.
Point 2: Add latest references in the introduction section. The introduction section needs a comparison of fluorinated compounds.
Response 2: Thanks for your reminding. We have revised the introduction to supplement the most recent references, while introducing different formulations of fluoropolymer-based reactive materials., references [3-9] are newly added. The specific modifications are as follows:
Fluoropolymer-based reactive composites are solid energetic materials mixed with reactive metal powder, alloy powder or intermetallic compound in polymer powder (typically PTFE), which can release chemical energy under high dynamic load and high strain rate conditions. Due to its unique properties, reactive composites have high application value in military and civilian fields [1, 2]. The PTFE/Al composite material is a typical reactive material, which is prepared by uniformly filling Al particles into a PTFE matrix, and cold-pressing and sintering. This material has become a benchmark for studying the properties and applications of reactive materials. Recently, for PTFE/Al composites, the formulations and fabrications [3], impact initiation [4] and chemical reaction [5] have been studied. Especially in terms of formulation, by adding high-density W powder to the PTFE/Al composites [6], the strength can be improved; by adding metal powders or metal oxides, such as MoO3 [7], Bi2O3 [8], CuO [9], etc., to the PTFE/Al composites, the heat release and gas product amount can be improved.
[3] C Ge, W Maimaitituersun, Y X Dong, et al. A study on the mechanical properties and impact-induced initiation characteristics of brittle PTFE/Al/W reactive materials [J], Materials 2017 (10) (2017) 452.
[4] F Bin, L Yuchun, W Shuangzhang, et al. A crack-induced initiation mechanism of Al-PTFE under quasi-static compression and the influencing factors [J], Mater. Des. 108 (15) (2016) 411–417.
[5] Z Song, L Jinxu, Y. Min et al. Effects of multi-component co-addition on reaction characteristics and impact damage properties of reactive materials [J], Mater. Des. 153 (5) (2018) 1–8.
[6] T Sun, Y F Zheng, Y Yuan, et al. Impact-Initiation Sensitivity of High-Temperature PTFE-Al-W Reactive Materials. Crystals. 2022, 12(1), 30.
[7] J Y Huang, X Fang, S Z Wu, et al. Mechanical Response and Shear-Induced Initiation Properties of PTFE/Al/MoO3 Reactive Composites. Materials. 2018, 11, 1200.
[8] Y Yuan, B Q Geng, T Sun, et al. Impact-Induced Reaction Characteristic and the Enhanced Sensitivity of PTFE/Al/Bi2O3 Composites. Polymers. 2019, 11(12), 2049.
[9] L L Ding, J Y Zhou, W H Tang, et al. Impact Energy Release Characteristics of PTFE/Al/CuO Reactive Materials Measured by a New Energy Release Testing Device. Polymers 2019, 11, 149.
Point 3: It is strongly recommended to add a subsection, 'practical implications of this study,' outlining the challenges in the current research, future work, and recommendations, before the conclusion.
Response 3: Thanks a lot for your reminding and comments. It is important and meaningful to explain to readers the challenges in the research, future work and recommendations. It should be noted to reviewers that this manuscript is a phased achievement. At present, we are carrying out the secondary development of AUTODYN finite element analysis software. Based on SPH algorithm, the state of any particle in the process of reactive composite jet forming will be tracked, and high-temperature particles will be screened out. Thereby, the composite material in the high temperature region can be replaced to realize the control of the reaction time of the reactive composite jet.
However, due to the short content of this part, to unify the layout of the article. Instead of adding a subsection, we have added this content to the conclusion, which we hope the reviewer will understand and agree. The specific modifications are as follows:
(d) In this paper, the distribution of density, temperature and velocity field in PTFE/Al reactive composite jet forming is only analyzed in the form of distribution diagram and observation points, and the state of all SPH particles cannot be monitored yet. In the future work, we will carry out the secondary development of AUTODYN finite element analysis software. Based on SPH algorithm, the state of any particle in the process of reactive composite jet forming will be tracked, and high-temperature particles will be screened out. Therefore, the PTFE/Al composite material in the high temperature region can be replaced to realize the control of the reaction time of the reactive composite jet.
Point 4: The title should be changed.
Response 4: Thanks a lot for your valuable comments, we now change the title of the article to ‘Formation behavior and reaction characteristic of PTFE/Al reactive jet’.
In addition, according to the comments of the reviewer, we have revised and improved the English of the article, and displayed it in revision mode in the revised manuscript. This part of the revision does not involve the content and data of the article.
Thank you again for your serious and careful work. I wish you success in your work and a happy life.

Round 2
Reviewer 1 Report
Аuthors cite themselves a lot: 6,10,11,13,14,19,20. Line 385 need to split the words Thejet.To understand research, there must be an end goal to it
Author Response
Response to the comments from reviewer 1 (Round 2)
Dear reviewer:
Thanks again to the reviewer and the Editorial Office for their feedback on our paper. Thank you very much for your recognition of the last revision and your careful review again. The comments have given very valuable and helpful advices for improving this article. We have addressed all issues mentioned in the reviewer's comments carefully and replied them point by point. The changes made in the revised manuscript have been highlighted with a green background. We hope the revised manuscript can meet the journal’s desired standard. We look forward to learning your response to our submission.
Yours sincerely,
Point 1: Аuthors cite themselves a lot: 6,10,11,13,14,19,20.
Response 1: Thanks a lot for your valuable reminder. As one of the main works of our laboratory is the research on the reactive shaped charge warheads, many of our results have been included in the references. Based on the reviewer's comment, we removed the references [6,11,13,20]. Since these references belong to the research background introduction, the deletion will not affect the research content of the manuscript. The specific revisions are in the revised manuscript.
Point 2: Line 385 need to split the words Thejet.
Response 2: Thanks a lot for your reminding. We are very sorry for our carelessness. We have carefully checked and corrected the mistakes in the manuscript.
Point 3: To understand research, there must be an end goal to it.
Response 3: Thanks a lot for your valuable comments. As suggested by the reviewer, it is very important to clarify the research objectives. Therefore, we have modified the first sentence of the abstract to clarify the research purpose of the manuscript. The specific modifications are as follows.
Abstract: To reveal the expansion phenomenon and reaction characteristics of aluminum particle filled polytetrafluoroethylene (PTFE/Al) reactive jet during the forming process, and to control the penetration and explosion coupling damage ability of the reactive jet, the temperature and density distribution of reactive jet were investigated by combining numerical simulation and experimental study.
We also revise the last paragraph of the introduction to highlight the research purpose. The specific modifications are as follows.
Different from the traditional metal jet, the expansion phenomenon and reaction mechanism of the reactive jet during the forming process are still unclear. These phenomena will greatly affect its damage ability to armor and concrete targets. Therefore, it is necessary to study the reactive jet forming process and reaction characteristic. Here, we began with the numerical simulation of reactive composite jet formation behaviors, the density and temperature field changes of the reactive composite jet during the formation process were analyzed.
Thank you again for your serious and careful work. I wish you success in your work and a happy life!
